# Universal Approximation of Residual Flows in Maximum Mean Discrepancy

**Zhifeng Kong** [1]  **Kamalika Chaudhuri** [1]

## Abstract

Normalizing flows are a class of flexible deep generative models that offer easy likelihood computation. Despite their empirical success, there is little theoretical understanding of their expressiveness. In this work, we study residual flows, a class of normalizing flows composed of Lipschitz residual blocks. We prove residual flows are universal approximators in maximum mean discrepancy. We provide upper bounds on the number of residual blocks to achieve approximation under different assumptions.

## 1. Introduction

Normalizing flows are a class of generative models that learn an invertible function to transform a predefined source distribution into a complex target distribution (Tabak et al., 2010; Tabak and Turner, 2013; Rezende and Mohamed, 2015). One category of normalizing flows called residual flows use residual networks (He et al., 2016) to construct the transformation (Rezende and Mohamed, 2015; Van Den Berg et al., 2018; Behrmann et al., 2019; Chen et al., 2019). These models have shown great success in complicated real-world tasks.

However, to ensure invertibility, these models apply additional Lipschitz constraints to each residual block. Under these strong constraints, how expressive these models are remains an open question. Formally, can they approximate certain target distributions to within any small error?

In this paper, we carry out a theoretical analysis on the expressive power of residual flows. We prove there exists a residual flow $F$ that achieves universal approximation in the mean maximum discrepancy (MMD, (Gretton et al., 2012)) metric. Formally, given a target distribution, we provide upper bounds on the number of residual blocks in $F$ such

that applying $F$ over the source distribution can approximate the target distribution in squared MMD (see (4)).

Although residual networks are universal approximators (Lin and Jegelka, 2018), the proof of approximation uses a non-invertible construction and therefore does not apply to residual flows. This reflects the main difficulty in analyzing residual flows: under strong Lipschitz and invertibility constraints, they become a very restricted function class. As an illustration, take the set of piecewise constant functions. Classical real analysis shows that piecewise constant functions can approximate any Lebesgue-integrable function and therefore any probability density. However, the invertible subset of all piecewise constant functions is the empty set! Consequently, this universal approximation result does not apply to normalizing flows. This difficulty leads to many negative results for normalizing flows: they are either unable to express or find it hard to approximate certain functions (Zhang et al., 2019; Koehler et al., 2020; Kong and Chaudhuri, 2020).

To tackle this problem, we adopt a new construction that satisfies the strong Lipschitz constraints in Behrmann et al. (2019). Specifically, we construct the residual blocks by multiplying a small $\epsilon$ to a pre-specified Lipschitz function. Therefore, as long as $\epsilon$ is small enough, the strong Lipschitz constraints are satisfied. We then analyze the following quantity: how much can the MMD be reduced if a new residual block is appended? Since this quantity is a function of $\epsilon$, we can analyze its Taylor expansion. With a first-order analysis and under mild conditions, we show there is an $F$ with $\Theta\left(\frac{1}{\delta}\left(\log\frac{1}{\delta}\right)^2\right)$ residual blocks that achieves (4) (see **Theorem** 1), where $\delta$ is the ratio between the final squared MMD and the initial squared MMD. With a second-order analysis and under more conditions, we show there is a shallower $F$ with only $\Theta\left(\log\frac{1}{\delta}\right)$ residual blocks that achieves (4) (see **Theorem** 2).

To sum up, we show residual flows are universal approximators in MMD under certain assumptions and provide explicit bounds on the number of residual blocks.

## 2. Related Work

The classic universal approximation theory for fully connected or residual neural networks in the function space are

[1]Department of Computer Science and Engineering, University of California San Diego, CA, USA. Correspondence to: Zhifeng Kong <z4kong@eng.ucsd.edu>, Kamalika Chaudhuri <kamalika@cs.ucsd.edu>.

Third workshop on *Invertible Neural Networks, Normalizing Flows, and Explicit Likelihood Models* (ICML 2021). Copyright 2021 by the author(s).

widely studied (Cybenko, 1989; Hornik et al., 1989; Hornik, 1991; Montufar et al., 2014; Telgarsky, 2015; Lu et al., 2017; Hanin, 2017; Raghu et al., 2017; Lin and Jegelka, 2018). However, these results do not generalize to residual flows (Rezende and Mohamed, 2015; Van Den Berg et al., 2018; Behrmann et al., 2019; Chen et al., 2019) for two reasons. First, the approximation theory for normalizing flows analyzes how well they can transform between *distributions*, rather than their ability to approximate a target function in the *function space*. Despite that $L^p$ universality in the function space may lead to distributional universality for triangular flows (Teshima et al., 2020), there is no similar results for non-triangular flows including residual flows. Second, the classic results do not consider the invertibility or the Lipschitz constraints of the neural networks, which greatly restrict the expressive power.

There are also universal approximation results for Lipschitz networks (Anil et al., 2019; Cohen et al., 2019; Tanielian et al., 2020). These results are related because in this work, we assume the expressive power of each Lipschitz residual block is large. However, these results only apply to functions defined on compact sets. Because compact sets are bounded, it is "easier" to satisfy the Lipschitz constraints. It is not trivial to extend their results to functions defined on $\mathbb{R}^d$.

Concerning the expressive power of generative networks, there are prior works showing feed-forward generator networks can approximate certain distributions (Lee et al., 2017; Bailey and Telgarsky, 2018; Lu and Lu, 2020; Perekrestenko et al., 2020). However, the results are again based on non-invertible constructions, so they do not apply to normalizing flows.

In the literature of normalizing flows, there are universal approximation results for several models including autoregressive flows (Germain et al., 2015; Kingma et al., 2016; Papamakarios et al., 2017; Huang et al., 2018; Jaini et al., 2019), coupling flows (Teshima et al., 2020; Koehler et al., 2020), and augmented normalizing flows (Zhang et al., 2019; Huang et al., 2020) [1]. There is also a continuous-time generalization of normalizing flows called neural ODEs (Chen et al., 2018; Dupont et al., 2019) with a universal approximation result (Zhang et al., 2019). We do not consider these flows in this paper. In addition, (Müller, 2020) suggests residual networks can approximate neural ODEs, but the invertibility is again not considered in this case.

On the expressive power of residual flows, all existing theoretical analysis present negative results for these models (Zhang et al., 2019; Koehler et al., 2020; Kong and Chaudhuri, 2020). These results indicate residual flows are either unable to express certain functions, or unable to approx-

imate certain distributions even with large depths. Compared to these results, our paper presents positive results for standard residual flows: given a source distribution $q$, they can approximate a target distribution $p$ in the MMD metric (Gretton et al., 2012) under certain conditions. We provide explicit upper bounds on the number of residual blocks (see **Theorem** 1 and **Theorem** 2).

## 3. Preliminaries

We first define the maximum mean discrepancy (MMD) metric between distributions below.

**Definition 1** (MMD, (Gretton et al., 2012)). *Let $q, p$ be two distributions on $\mathbb{R}^d$. Then,*

$$\text{MMD}(q,p)^2 = \begin{aligned} &\mathbb{E}_{z,z'\sim q}K(z,z') + \mathbb{E}_{x,x'\sim p}K(x,x') \\ &-2\cdot\mathbb{E}_{z\sim q,x\sim p}K(z,x) \end{aligned} \tag{1}$$

*for some kernel function $K(\cdot,\cdot)$. Let $\phi : \mathbb{R}^d \to \mathbb{R}^{d_\phi}$ be the feature map associated with $K$: $K(x,z) = \phi(x)^\top \phi(z)$, where we assume $d_\phi < \infty$. Then, the squared MMD can be simplified as*

$$\text{MMD}(q,p)^2 = \|\mathbb{E}_{z\sim q}\phi(z) - \mathbb{E}_{x\sim p}\phi(x)\|_2^2. \tag{2}$$

Next, we define a residual flow as a composition of invertible layers parameterized as $\mathbf{Id} + f$, where $\mathbf{Id}$ is the identity map and $f$ is $\frac{1}{2}$-Lipschitz[2]. The class of residual flows include planar flows (Rezende and Mohamed, 2015), Sylvester flows (Van Den Berg et al., 2018), and the more general invertible residual networks (Behrmann et al., 2019; Chen et al., 2019). In these models every $f_i$ is parameterized as a certain kind of fully-connected neural network. Since the expressive power of $(\frac{1}{2}-)$Lipschitz neural networks on $\mathbb{R}^d$ remains an open problem, in this paper we assume every $f_i$ can be selected as any $\frac{1}{2}$-Lipschitz function. Formally, we make the following definition.

**Definition 2** (Residual flows). *The set of $N$-block residual flows is defined as*

$$\mathcal{F}_N = \begin{aligned} \{&(\mathbf{Id} + f_N) \circ \cdots \circ (\mathbf{Id} + f_1) : \\ &\text{each } f_i \text{ is } \tfrac{1}{2}\text{-Lipschitz}\} . \end{aligned} \tag{3}$$

Now we state the main problem. Let $q_{\text{source}}$ and $p_{\text{target}}$ be two distributions on $\mathbb{R}^d$, where $q_{\text{source}}$ is the source distribution and $p_{\text{target}}$ is the target distribution. We aim to answer the following problem in this paper.

**Problem Statement.** *Let $\delta > 0$ be a small number. For any pair of distributions $q_{\text{source}}$ and $p_{\text{target}}$ on $\mathbb{R}^d$ satisfying*

---

[1]In an augmented normalizing flow, there is an auxiliary random variable concatenated with the data, so the transformations operate on a higher dimensional space.

[2]According to the fixed-point theorem, $\mathbf{Id} + f$ is invertible as long as the Lipschitz constant of $f$ is strictly less than 1. For algebraic convenience, we restrict the Lipschitz constant to be at most $\frac{1}{2}$.

$\mathrm{MMD}(q_{\text{source}}, p_{\text{target}}) < \infty$, *does there exist an $N$ and $F \in \mathcal{F}_N$ such that*

$$\mathrm{MMD}(F\#q_{\text{source}}, p_{\text{target}})^2 \leq \delta \cdot \mathrm{MMD}(q_{\text{source}}, p_{\text{target}})^2, \tag{4}$$

*where $F\#q$ refers to the distribution obtained by applying $F$ over $q$?*

In this paper, we prove existence of such $F$ with a loose bound on $N$ using first-order analysis under mild assumptions (see Section 4), and provide a tighter bound on $N$ using second-order analysis under more assumptions (see Section 5).

## 4. A Bound with First-Order Analysis

In this section, we show under mild conditions, there exists a residual flow $F$ with $N = \Theta\left(\frac{1}{\delta}\left(\log\frac{1}{\delta}\right)^2\right)$ residual blocks that achieves (4). The idea is to show that a single residual block can reduce the squared MMD by a certain fraction, so $F$ is obtained by stacking an enough number of these residual blocks. To begin with, we make the follow definition.

**Definition 3.** *For distributions $q$, $p$, and a $\frac{1}{2}$-Lipschitz function $f$, we define the improvement of the squared MMD by $\mathbf{Id} + f$ as*

$$\Delta(q, p; f) = \mathrm{MMD}(q, p)^2 - \mathrm{MMD}((\mathbf{Id}+f)\#q, p)^2. \tag{5}$$

Then, if $\Delta(q, p; f) > 0$, the residual block $\mathbf{Id} + f$ is helpful for reducing the squared MMD. It is straightforward to see that $\sup\{\Delta(q, p; f): f \text{ is } \frac{1}{2}\text{-Lipschitz}\} \geq 0$. In order to construct an $f$ that has a large $\Delta(q, p; f)$, we choose $f = \hat{f}_\epsilon$ defined below.

**Definition 4.** *Define $\psi(p, q) = (\mathbb{E}_{x \sim p} - \mathbb{E}_{x \sim q})\phi(x)$, $g(z) = \psi(p,q)^\top \phi(z)$, and $\hat{f}_\epsilon = \epsilon \cdot \nabla g$, where $\epsilon > 0$. Then, $\mathrm{MMD}(q, p) = \|\psi(p, q)\|$. In addition, $\hat{f}_\epsilon(z) = \epsilon J_\phi(z)\psi(p, q)$, where $J_\phi$ is the Jacobian matrix of $\phi$.*

Then, according to (2) and (5),

$$\begin{aligned}
\Delta(q, p; \hat{f}_\epsilon) = \quad & \mathbb{E}_{z \sim q, x \sim q}\phi(z)^\top \phi(x) \\
& - \mathbb{E}_{z \sim q, x \sim q}\phi(z + \hat{f}_\epsilon(z))^\top \phi(x + \hat{f}_\epsilon(x)) \\
& + 2 \cdot \mathbb{E}_{z \sim q, x \sim p}\phi(z + \hat{f}_\epsilon(z))^\top \phi(x) \\
& - 2 \cdot \mathbb{E}_{z \sim q, x \sim p}\phi(z)^\top \phi(x).
\end{aligned} \tag{6}$$

Note that $\Delta(q, p; \hat{f}_\epsilon)$ is a function of $\epsilon$. We then analyze the first-order Taylor expansion of $\Delta(q, p; \hat{f}_\epsilon)$ at $\epsilon = 0^+$, denoted as $\Delta_1(q, p; \hat{f}_\epsilon)$. Then, $\Delta(q, p; \hat{f}_\epsilon) = \Delta_1(q, p; \hat{f}_\epsilon) + \mathcal{O}\left(\epsilon^2\right)$. With some arithmetic, we have

$$\Delta_1(q, p; \hat{f}_\epsilon) = 2\psi(p, q)^\top \mathbb{E}_{z \sim q}\phi(z + \hat{f}_\epsilon(z)). \tag{7}$$

We have the following bound on $\Delta_1(q, p; \hat{f}_\epsilon)$.

**Lemma 1.** *If $d_\phi < \infty$, and the minimum singular value $\sigma_{\min}(J_\phi(z)) \geq \sqrt{b} > 0$ holds for any $z \in \mathbb{R}^d$, then*

$$\Delta_1(q, p; \hat{f}_\epsilon) \geq 2\epsilon b \cdot \mathrm{MMD}(q, p)^2. \tag{8}$$

Since $\Delta(q, p; \hat{f}_\epsilon) = \Delta_1(q, p; \hat{f}_\epsilon) + \mathcal{O}\left(\epsilon^2\right)$, when $\epsilon$ is small, the residual block $\mathbf{Id} + \hat{f}_\epsilon$ can indeed reduce the squared MMD by a certain fraction ($\geq 2\epsilon b$). Next, as we require $f = \hat{f}_\epsilon$ to be $\frac{1}{2}$-Lipschitz, we show under certain conditions the Lipschitz constant of $\hat{f}_\epsilon$ is $\mathcal{O}(\epsilon)$ in the following lemma.

**Lemma 2.** *If for any $z \in \mathbb{R}^d$, the Lipschitz constant of each element in $J_\phi(z)$ is no more than a universal constant $L_{\text{Jac}}$, then*

$$\mathrm{Lip}(\hat{f}_\epsilon) \leq \sqrt{d \cdot d_\phi} L_{\text{Jac}} \mathrm{MMD}(q, p) \cdot \epsilon. \tag{9}$$

With these tools, we can construct an $F \in \mathcal{F}_N$ that achieves (4) in the following theorem.

**Theorem 1.** *Under the conditions of **Lemma 1** and **Lemma 2**, there exists an $F \in \mathcal{F}_N$ with $N = \Theta\left(\frac{1}{\delta}\left(\log\frac{1}{\delta}\right)^2\right)$ that achieves (4).*

The proof is deferred to Section A.3. The main idea in the proof is to construct each $f_i$ iteratively based on $f_1$ through $f_{i-1}$, so that adding this residual block can reduce the squared MMD by a certain fraction as indicated in **Lemma 1**. The bound is obtained by carefully balancing $\epsilon$, $\delta$, and $N$.

## 5. A Tighter Bound with Second-Order Analysis

In this section, we show under a few additional assumptions, there exists a much smaller $N = \mathcal{O}\left(\log\frac{1}{\delta}\right)$ and $F \in \mathcal{F}_N$ such that $F$ achieves (4). The idea is to bound the second-order remainder of the Taylor expansion of $\Delta(q, p; \hat{f}_\epsilon)$: $\Delta_2(q, p; \hat{f}_\epsilon) = \Delta(q, p; \hat{f}_\epsilon) - \Delta_1(q, p; \hat{f}_\epsilon) = \mathcal{O}\left(\epsilon^2\right)$. Once $\Delta_2(q, p; \hat{f}_\epsilon)$ is explicitly bounded we can pick a small constant $\epsilon$ for every residual block [3] so $\Delta(q, p; \hat{f}_\epsilon)$ is lower bounded by a universal constant times $\mathrm{MMD}(q, p)^2$. This then yields the $\mathcal{O}\left(\log\frac{1}{\delta}\right)$ bound for $N$. Now, we provide an explicit bound on $\Delta_2(q, p; \hat{f}_\epsilon)$ in the following lemma.

**Lemma 3.** *Let $B, C, L_{\text{feat}}$ be positive constants. If for any $z \in \mathbb{R}^d$, the maximum singular value $\sigma_{\max}(J_\phi(z)) \leq \sqrt{B}$, the absolute value of any eigenvalue $|\lambda(\nabla^2\phi_i(z))| \leq C$ for any $1 \leq i \leq d_\phi$, and $\phi$ is $L_{\text{feat}}$-Lipschitz, then*

$$\begin{aligned}
|\Delta_2(q, p; \hat{f}_\epsilon)| \quad &\leq \epsilon^2 \cdot \mathrm{MMD}(q, p)^2 \cdot B \cdot \Big(B + \\
& \|\psi(p, q)\|\sqrt{d_\phi}C(1 + \epsilon L_{\text{feat}}\sqrt{B})\Big).
\end{aligned} \tag{10}$$

---

[3] In **Theorem 1**, the $\epsilon$ for each residual block is related to $\delta$ in order to eliminate the effect by the unknown second-order terms. Here $\epsilon$ is independent with $\delta$.

Given this explicit bound on $\Delta_2(q, p; \hat{f}_\epsilon)$, we can pick a small $\epsilon$ such that $|\Delta_2(q, p; \hat{f}_\epsilon)| \leq \frac{1}{2}\Delta_1(q, p; \hat{f}_\epsilon)$ so that $\Delta(q, p; \hat{f}_\epsilon) \geq \frac{1}{2}\Delta_1(q, p; \hat{f}_\epsilon)$. Once this lower bound on $\Delta(q, p; \hat{f}_\epsilon)$ is achieved, the squared MMD is multiplied by at most a universal constant less than 1 when the new residual block $\mathbf{Id} + \hat{f}_\epsilon$ is added. We formalize the result in the following theorem.

**Theorem 2.** *Under the conditions of **Lemma 1**, **Lemma 2**, and **Lemma 3**, there exists an $F \in \mathcal{F}_N$ with $N = \Theta\left(\log\frac{1}{\delta}\right)$ that achieves* (4).

The proof is deferred to Section A.5. The main idea in the proof is to construct each $f_i$ in a similar way as in **Theorem 1**, but $\epsilon$ is selected as a universal constant according to **Lemma 3**.

## 6. Conclusions

Normalizing flows are a class of flexible deep generative models that offers easy likelihood computation. Despite their empirical success, there is little theoretical understanding on whether they are universal approximators in transforming between probability distributions. In this work, we prove residual flows are indeed universal approximators in maximum mean discrepancy. Upper bounds on the number of residual blocks to achieve approximation are provided. Under mild conditions, we show $\Theta\left(\frac{1}{\delta}\left(\log\frac{1}{\delta}\right)^2\right)$ residual blocks can achieve (4) (see **Theorem 1**). Under more conditions, we show as few as $\Theta\left(\log\frac{1}{\delta}\right)$ residual blocks can achieve (4) (see **Theorem 2**).

There are a large number of open problems. One extension is to build universal approximation theory for residual flows in more general probability metrics such as the integral probability metrics (Müller, 1997) and the $f$-divergences (Csiszár and Shields, 2004). Another direction is to extend the proposed universal approximation theory to other classes of normalizing flows such as autoregressive flows. A final open problem is to look at normalizing flows in real-world applications, and analyze their expressive power under practical assumptions.

## Acknowledgement

We thank Songbai Yan for helpful feedback.

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

## A. Omitted Proofs

### A.1. Proof of Lemma 1

*Proof.* According to (7) and the chain rule,

$$
\begin{aligned}
\Delta_1(q, p; \hat{f}_\epsilon) &= 2 \cdot \mathbb{E}_{z \sim q} \left( \psi(p, q)^\top J_\phi(z)^\top \hat{f}_\epsilon(z) \right) \\
&= 2\epsilon \cdot \mathbb{E}_{z \sim q} \left( \psi(p, q)^\top J_\phi(z)^\top \nabla g(z) \right) \\
&= 2\epsilon \cdot \mathbb{E}_{z \sim q} \left( \psi(p, q)^\top J_\phi(z)^\top J_\phi(z) \psi(p, q) \right) \\
&\geq 2\epsilon \cdot \min_{z \in \mathbb{R}^d} \left( \psi(p, q)^\top J_\phi(z)^\top J_\phi(z) \psi(p, q) \right) \\
\left( \mathrm{MMD}(q, p)^2 = \| \psi(p, q) \|^2 \right) \quad &\geq 2\epsilon \cdot \mathrm{MMD}(q, p)^2 \min_{z \in \mathbb{R}^d} \lambda_{\min} \left( J_\phi(z)^\top J_\phi(z) \right) \\
&\geq 2\epsilon \cdot \mathrm{MMD}(q, p)^2 \min_{z \in \mathbb{R}^d} \sigma_{\min}^2(J_\phi(z)) \\
&\geq 2\epsilon b \cdot \mathrm{MMD}(q, p)^2.
\end{aligned}
$$

$\square$

### A.2. Proof of Lemma 2

*Proof.* For any $x, y \in \mathbb{R}^d$,

$$
\begin{aligned}
\frac{\| \hat{f}_\epsilon(y) - \hat{f}_\epsilon(x) \|}{\| y - x \|} &= \frac{\epsilon \sqrt{\sum_{i=1}^d \left( \sum_{j=1}^{d_\phi} (J_\phi(y) - J_\phi(x))_{ij} \psi(p, q)_j \right)^2}}{\| y - x \|} \\
&\leq \frac{\epsilon \sqrt{\sum_{i=1}^d \left( \sum_{j=1}^{d_\phi} L_{\mathrm{Jac}} \| y - x \| \psi(p, q)_j \right)^2}}{\| y - x \|} \\
&\leq \epsilon \sqrt{d} L_{\mathrm{Jac}} \| \psi(p, q) \|_1 \\
&\leq \epsilon \sqrt{d \cdot d_\phi} L_{\mathrm{Jac}} \| \psi(p, q) \|_2 \\
&= \epsilon \sqrt{d \cdot d_\phi} L_{\mathrm{Jac}} \mathrm{MMD}(q, p).
\end{aligned}
$$

Therefore, by taking the supreme over the left-hand-side, we have the Lipschitz constant of $\hat{f}_\epsilon$ is upper bounded by the right-hand-side. $\square$

### A.3. Proof of Theorem 1

*Proof.* Let $r > 0$ and $\epsilon = r/N$. Define

$$
D_n(r) = \mathrm{MMD}((\mathbf{Id} + f_n) \circ \cdots \circ (\mathbf{Id} + f_1) \# q_{\mathrm{source}}, p_{\mathrm{target}})^2
$$

where each

$$
f_i(z) = \epsilon J_\phi(z) \psi(p_{\mathrm{target}}, (\mathbf{Id} + f_{i-1}) \circ \cdots \circ (\mathbf{Id} + f_1) \# q_{\mathrm{source}}).
$$

Note that each $f_i$ is exactly the $\hat{f}_\epsilon$ in **Definition** 4 for $q = (\mathbf{Id} + f_{i-1}) \circ \cdots \circ (\mathbf{Id} + f_1) \# q_{\mathrm{source}}$ and $p = p_{\mathrm{target}}$.

By **Lemma** 1,

$$
D_n(r) \leq \left( 1 - 2b \frac{r}{N} + \mathcal{O}\left( \frac{r^2}{N^2} \right) \right) D_{n-1}(r).
$$

Therefore,

$$
\begin{aligned}
D_N(r) &\leq \prod_{n=1}^N \left( 1 - 2b \frac{r}{N} + \mathcal{O}\left( \frac{r^2}{N^2} \right) \right) \mathrm{MMD}(q_{\mathrm{source}}, p_{\mathrm{target}})^2 \\
&\leq \left( e^{-2br} + \mathcal{O}\left( \frac{r^2}{N} \right) \right) \mathrm{MMD}(q_{\mathrm{source}}, p_{\mathrm{target}})^2.
\end{aligned}
$$

For small $\delta > 0$, we choose

$$
r = \frac{1}{2b} \log \frac{2}{\delta}, \ N = \Theta\left( \frac{r^2}{\delta} \right) = \Theta\left( \frac{1}{\delta} \left( \log \frac{1}{\delta} \right)^2 \right).
$$

Then, we have
$$D_N(r) \leq \delta \cdot \text{MMD}(q_{\text{source}}, p_{\text{target}})^2.$$

Note that
$$\epsilon = \frac{r}{N} = \Theta\left(\frac{\delta}{r}\right) = \Theta\left(\frac{\delta}{\log \frac{1}{\delta}}\right).$$

Therefore, by **Lemma** 2, when $\delta$ is small enough, the Lipschitz constant of each $f_n$ is less than $\frac{1}{2}$. $\qquad\square$

### A.4. Proof of Lemma 3

*Proof.* According to (1) and (5),
$$\Delta(q, p; \hat{f}_\epsilon) = \mathbb{E}_{z \sim q, x \sim q}(K(z, x) - K(z + \hat{f}_\epsilon(z), x + \hat{f}_\epsilon(x))) + 2\mathbb{E}_{z \sim q, x \sim p}(K(z + \hat{f}_\epsilon(z), x) - K(z, x))$$

There is a closed-form expression for $\Delta_2(q, p; \hat{f}_\epsilon)$. According to the remainder of multivariate Taylor polynomials, there exist two maps $\xi_1, \xi_2 : \mathbb{R}^d \to (0, 1)$ such that

$$
\begin{aligned}
\Delta_2(q, p; \hat{f}_\epsilon) = \quad & \mathbb{E}_{z \sim q}\mathbb{E}_{x \sim p}\hat{f}_\epsilon(z)^\top[\nabla_{zz}^2 K(z + \xi_1(z)\hat{f}_\epsilon(z), x + \xi_1(x)\hat{f}_\epsilon(x))]\hat{f}_\epsilon(z) && (=: \Delta_2^{(1)}) \\
& -\mathbb{E}_{z \sim q}\mathbb{E}_{x \sim q}\hat{f}_\epsilon(z)^\top[\nabla_{zz}^2 K(z + \xi_1(z)\hat{f}_\epsilon(z), x + \xi_1(x)\hat{f}_\epsilon(x))]\hat{f}_\epsilon(z) && (=: -\Delta_2^{(2)}) \\
& -\mathbb{E}_{z \sim q}\mathbb{E}_{x \sim q}\hat{f}_\epsilon(z)^\top[\nabla_{zx}^2 K(z + \xi_2(z)\hat{f}_\epsilon(z), x + \xi_2(x)\hat{f}_\epsilon(x))]\hat{f}_\epsilon(x) && (=: -\Delta_2^{(3)}) \\
= \quad & \Delta_2^{(1)} - \Delta_2^{(2)} - \Delta_2^{(3)}.
\end{aligned}
$$

First, we bound $|\Delta_2^{(1)} - \Delta_2^{(2)}|$. Define
$$\psi' := (\mathbb{E}_{x \sim p} - \mathbb{E}_{x \sim q})\phi(x + \xi_1(x)\hat{f}_\epsilon(x)) = \psi(p, q) + \hat{\psi}_\epsilon.$$

Since $\phi$ is $L_{\text{feat}}$-Lipschitz and $0 < \xi_1(x) < 1$, we have

$$
\begin{aligned}
\|\hat{\psi}_\epsilon\| \quad &\leq \quad \sup_{x \in \mathbb{R}^d} L_{\text{feat}}\xi_1(x)\|\hat{f}_\epsilon(x)\| \\
&\leq \quad \sup_{x \in \mathbb{R}^d} L_{\text{feat}}\epsilon\|J_\phi(x)\psi(p, q)\| \\
&\leq \quad \sup_{x \in \mathbb{R}^d} L_{\text{feat}}\epsilon\sigma_{\max}(J_\phi(x))\|\psi(p, q)\| \\
&\leq \quad \epsilon L_{\text{feat}}\sqrt{B}\|\psi(p, q)\|.
\end{aligned}
$$

Therefore,
$$\|\psi'\| \leq (1 + \epsilon L_{\text{feat}}\sqrt{B})\|\psi(p, q)\|.$$

For any $z', v \in \mathbb{R}^d$,

$$
\begin{aligned}
\left|v^\top[\nabla_{zz}^2(\phi(z')^\top\psi')]v\right| \quad &= \quad \left|\sum_{i=1}^{d_\phi} \psi_i' v^\top \nabla^2 \phi_i(z') v\right| \\
&\leq \quad \|\psi'\|\|v\|^2 \sqrt{\sum_{i=1}^{d_\phi} \max \lambda(\nabla^2 \phi_i(z'))^2} \\
&\leq \quad \sqrt{d_\phi} C \|\psi'\|\|v\|^2.
\end{aligned}
$$

By letting $z' = z + \xi_1(z)\hat{f}_\epsilon(z)$ and $v = \hat{f}_\epsilon(z)$, we have

$$
\begin{aligned}
|\Delta_2^{(1)} - \Delta_2^{(2)}| \quad &= \quad \left|\mathbb{E}_{z \sim q}v^\top[\nabla_{zz}^2(\phi(z')^\top\psi')]v\right| \\
&\leq \quad \epsilon^2\|\psi(p, q)\|^2\sigma_{\max}^2(J_\phi)\sqrt{d_\phi}C\|\psi'\| \\
&\leq \quad \epsilon^2\|\psi(p, q)\|^3\sqrt{d_\phi}BC(1 + \epsilon L_{\text{feat}}\sqrt{B}).
\end{aligned}
$$

Next, we bound $\Delta_2^3$. Observe that

$$\Delta_2^{(3)} = -\mathbb{E}_{z \sim q}\mathbb{E}_{x \sim q}\hat{f}_\epsilon(z)^\top J_\phi(z + \xi_2(z)\hat{f}_\epsilon(z))J_\phi(x + \xi_2(x)\hat{f}_\epsilon(x))^\top\hat{f}_\epsilon(x).$$

Therefore,

$$\begin{aligned}
|\Delta_2^{(3)}| \quad &\leq \max_{z \in \mathbb{R}^d} \|\hat{f}_\epsilon(z)\|^2 \max_{z \in \mathbb{R}^d} \|J_\phi(z)\|^2 \\
&\leq \epsilon^2 \|\psi(p,q)\|^2 \max_{z \in \mathbb{R}^d} \sigma_{\max}^4(J_\phi(z)) \\
&\leq \epsilon^2 \|\psi(p,q)\|^2 B^2.
\end{aligned}$$

Combining these bounds, we have

$$|\Delta_2(q,p;\hat{f}_\epsilon)| \leq \epsilon^2 \cdot \mathrm{MMD}(q,p)^2 \left( \|\psi(p,q)\| \sqrt{d_\phi} BC(1 + \epsilon L_{\mathrm{feat}} \sqrt{B}) + B^2 \right).$$

$\square$

### A.5. Proof of Theorem 2

*Proof.* Let

$$q_0 = q_{\mathrm{source}} \text{ and } q_m = (\mathbf{Id} + f_m) \circ \cdots \circ (\mathbf{Id} + f_1) \# q_{\mathrm{source}},$$

where each

$$f_i(z) = \epsilon J_\phi(z) \psi(p_{\mathrm{target}}, q_{i-1}).$$

Define $\psi_0 = \psi(p_{\mathrm{target}}, q_0)$ and assume $\|\psi(p_{\mathrm{target}}, q_m)\| \leq \|\psi_0\|$ (which we will prove by induction). Note that

$$\Delta(q_m, p_{\mathrm{target}}; f_m) = \mathrm{MMD}(q_m, p_{\mathrm{target}})^2 - \mathrm{MMD}(q_{m+1}, p_{\mathrm{target}})^2.$$

According to **Lemma** 1 and **Lemma** 3, we have

$$\begin{aligned}
\frac{\Delta(q_m, p_{\mathrm{target}}; f_m)}{\mathrm{MMD}(q_m, p_{\mathrm{target}})^2} \quad &\geq 2b\epsilon - \left( \|\psi(p_{\mathrm{target}}, q_m)\| \sqrt{d_\phi} BC + B^2 \right) \epsilon^2 - \|\psi(p_{\mathrm{target}}, q_m)\| \sqrt{d_\phi} B^{\frac{3}{2}} CL_{\mathrm{feat}} \epsilon^3 \\
&\geq 2b\epsilon - \left( \|\psi_0\| \sqrt{d_\phi} BC + B^2 \right) \epsilon^2 - \|\psi_0\| \sqrt{d_\phi} B^{\frac{3}{2}} CL_{\mathrm{feat}} \epsilon^3
\end{aligned}$$

When

$$\epsilon \leq \epsilon_\Delta = \min \left( \frac{b}{2 \left( \|\psi_0\| \sqrt{d_\phi} BC + B^2 \right)}, \sqrt{\frac{b}{2\|\psi_0\| \sqrt{d_\phi} B^{\frac{3}{2}} CL_{\mathrm{feat}}}} \right),$$

we have

$$\frac{\Delta(q_m, p_{\mathrm{target}}; f_m)}{\mathrm{MMD}(q_m, p_{\mathrm{target}})^2} \geq b\epsilon.$$

Next, by **Lemma** 2, in order to satisfy the Lipschitz condition, we require

$$\epsilon \leq \frac{1}{2\sqrt{d \cdot d_\phi} L_{\mathrm{Jac}} \|\psi(p_{\mathrm{target}}, q_m)\|}.$$

This is satisfied when we assign

$$\epsilon \leq \epsilon_{\mathrm{Lip}} := \frac{1}{2\sqrt{d \cdot d_\phi} L_{\mathrm{Jac}} \|\psi_0\|}.$$

Now, we set

$$\epsilon = \hat{\epsilon} := \min(\epsilon_\Delta, \epsilon_{\mathrm{Lip}}).$$

Then, we have

$$\mathrm{MMD}(q_{m+1}, p_{\mathrm{target}})^2 \leq (1 - b\hat{\epsilon}) \cdot \mathrm{MMD}(q_m, p_{\mathrm{target}})^2,$$

which also implies $\|\psi(p_{\mathrm{target}}, q_{m+1})\| \leq \sqrt{1 - b\hat{\epsilon}} \|\psi_0\| \leq \|\psi_0\|$. Finally, in order to satisfy (4), we only need to take the number of residual blocks as

$$N = \frac{\log \frac{1}{\delta}}{\log \frac{1}{1-b\hat{\epsilon}}}.$$

$\square$