# OpenReview forum: "Universal Approximation of Residual Flows in Maximum Mean Discrepancy"
_ICML.cc/2021/Workshop/INNF — INNF+ 2021 spotlighttalk_

### Official Review · Reviewer_qsBp · 2021-06-10

**Rating:** Accept
**Confidence:** 5

**Summary:**

This paper analyzes the expressiveness of Residual Flows in the MMD metric. Towards this end, the paper provides two bound on the depth of the network N based on first and second-order Taylor approximations of the delta reduction in the pushforward MMD metric under F. Overall, the paper provides a positive outlook on the expressiveness of Residual Flows.

**Justification For Rating:**

The paper is well written and I thought the main Lemma's in the proof to be elegant. I did not however check in detail the correctness of the results in the Appendix.

My main question/concern for the authors is that there are now numerous universality results for flows. They invoke different notions of universality, from modeling arbitrary functions in a class to distributional. Why is MMD universality a more compelling tool to analyze ResFlows? Does it align better with practice? I understand that the previous result by Zhang et. al 2019 on i-ResNet provides a negative example for a function that i-ResNets cannot hope to express (e.g. f(x) = -x) but they correct this by adding zero-padding. How does the theory presented in this paper inform practice?

Finally, it would be great if the authors could provide a concrete example with a few common kernels (e.g. Gaussian) of empirical approximation results.


Other Concerns:
- Statement/Claim on line 22 right should be refined: "Consequently, this universal approximation result does not
apply to normalizing flows." See Teshima et. al 2020 Lemma 1 or Appendix A where L_p universality in function space leads to distributional universality.
- Same as above for lines 60-63 left

Other questions to the authors:
- In footnote 2 can you expand on why restricting the Lipschitz constant to 1/2 is more desirable? If it's strictly <1 then the map is invertible. Why is approaching 1 considered an extreme case? Seems like this is mostly algebraically convenient.
- In Lemma 1, if \eps is small and \sqrt{b} is < \sigma_{min} then isn't the term 2\eps*b an extremely small reduction? Would this not suggest that the number of layers needed to reduce MMD is quite large?

---

### Official Review · Reviewer_g5Wj · 2021-06-12

**Rating:** Accept
**Confidence:** 3

**Summary:**

A nice paper deriving a way to approximate a target distribution using a residual flow from a source distribution to an arbitrary maximum mean discrepancy.

**Justification For Rating:**

The paper provides a theoretical analysis of expressive power of residual flows. The analysis is quite general, with relatively simple assumptions. I think the paper is a great fit for the workshop.

A question I had reading the paper: the analysis seems to provide a network  which is specific to the kernel K used in the MMD distance. How hard would it be to generalize the analysis to e.g. the maximum MMD distance over some class of kernels? And also how hard would it be to go to kernels with infinite-dimensional representations ф?

I would recommend adding an intuition for what the residual blocks that you derive, e.g. in lemma 1 are doing. It would also be great if you could add an experiment, using your construction and verifying your theoretical results empirically.

---

### Official Review · Reviewer_VztE · 2021-06-13

**Rating:** Accept
**Confidence:** 4

**Summary:**

The manuscript presents a proof that residual flows are universal approximators in the maximum mean discrepancy distance between distributions. It also provides two bounds (under two sets of conditions) on the number of residual flows required for a given approximation accuracy.

**Justification For Rating:**

This contribution makes an important step in understanding this class of normalizing flow building blocks, and as pointed out by the authors, shows a refreshingly positive result on the flexibility of residual flows. The paper is well-written, and I appreciated the "Related Work" section - it nicely places this work in the context of what's been done previously. The (partial) proofs in in the main text make sense to me.

*Minor comment*: I was not familiar with the notation used to define $\psi(p, q)$ in Definition 4.

---

### Decision · Program_Chairs · 2021-06-14

Accept (spotlight talk)